# Whole-Genome Sequencing of a Potentially Novel *Aeromonas* Species Isolated from Diseased Siberian Sturgeon (*Acipenser baerii*) Using Oxford Nanopore Sequencing

**DOI:** 10.3390/microorganisms13071680

**Published:** 2025-07-17

**Authors:** Akzhigit Mashzhan, Izat Smekenov, Serik Bakiyev, Kalamkas Utegenova, Diana Samatkyzy, Asset Daniyarov, Ulykbek Kairov, Dos Sarbassov, Amangeldy Bissenbaev

**Affiliations:** 1Scientific Research Institute of Biology and Biotechnology Problems, Al-Farabi Kazakh National University, Almaty 050040, Kazakhstan; 2Almaty Branch of the National Center for Biotechnology, Central Reference Laboratory, Zhahanger St. 14, Almaty 050054, Kazakhstan; 3Department of Molecular Biology and Genetics, Faculty of Biology and Biotechnology, Al-Farabi Kazakh National University, Almaty 050040, Kazakhstan; 4Educational Programs for Training Teachers of Biology, Geography, Chemistry, Faculty of Natural Geography, Makhambet Utemisov West Kazakhstan University, Uralsk 090009, Kazakhstan; 5Center for Life Sciences, National Laboratory Astana, Nazarbayev University, Astana 010000, Kazakhstan; 6Faculty of Natural Sciences, L.N. Gumilyev Eurasian National University, 2 Kanysh Satpayev Street, Astana 010008, Kazakhstan

**Keywords:** *Aeromonas* species, genome, taxonomy, *Ac. baerii*, average nucleotide identity, whole-genome sequencing, digital DNA–DNA hybridization, TYGS

## Abstract

*Aeromonas* spp. are opportunistic pathogens that are widely distributed in water sources, with several species being associated with fish and human diseases. We have previously identified an *Aeromonas* AB005 isolate from diseased *Acipencer baerii*. This isolate was identified as *A. hydrophila* based on the 16S rRNA and *gyrB* gene sequences. However, this novel strain does not produce indole and tested negative for ornithine decarboxylase and d-xylose fermentation—differences that set it apart from typical *A. hydrophila* strains. In the present study, this strain was subjected to whole-genome sequencing and compared with the genomes of the type strain (*Aeromonas hydrophila* ATCC 7966^T^) and other *Aeromonas* spp. Comprehensive genome analysis suggests that AB005 represents a distinct species within the genus. The draft genome of the AB005 strain comprises 4,780,815 base pairs with a GC content of 61.2% and contains 6104 predicted protein-coding sequences along with numerous genes implicated in antibiotic resistance. The core/pan-genome analysis reveals extensive genetic diversity, indicative of a dynamic genomic structure. These findings collectively underscore the taxonomic distinction of the AB005 strain as a novel species and highlight its potential pathogenic implications in aquaculture and public health settings.

## 1. Introduction

The genus *Aeromonas* is one of the four validly published genera within the family *Aeromonadaceae* [1]. *Aeromonas* spp. are opportunistic pathogens that can colonize and infect multiple hosts. To date, the genus includes 32 recognized species, with 19 implicated in human infections, including bacteremia, diarrhea, and wound infections. If not properly treated, these infections can progress to systemic levels, leading to septicemia [2,3,4,5]. Specifically, *A. veronii*, *A. caviae*, and *A. hydrophila* are associated with infections of soft tissue [6] and the hepatobiliary system, and they may also cause ocular, respiratory, joint, and bone infections, typically following an initial case of septicemia [7,8,9].

*Aeromonas* spp. are commonly found in marine environments, making fish and other seafood the typical sources for isolating these microorganisms. Consequently, they are recognized in aquaculture as potential pathogens that cause diseases such as furunculosis and septicemia. Additionally, *Aeromonas* can be isolated from other food sources, including vegetables, beef, pork, and dairy products [10,11].

These organisms are facultative anaerobic, Gram-negative, non-spore-forming coccobacilli or bacilli that are generally motile and capable of reducing nitrate to nitrite. They exhibit resistance to the vibriostatic agent O/129 (2,4-diamino-6,7-diisopropylpteridine) [12], are both oxidase- and catalase-positive, and can tolerate increasing NaCl concentrations from 0.3% to 5% [13]. Additionally, they are capable of fermenting glucose [14].

The taxonomy of the *Aeromonas* genus is complex; although each species appears to have unique phenotypic characteristics, biochemical identification is laborious and often imprecise, showing poor correlation with genotypic identification [15,16,17]. The 16S rRNA gene sequence similarity among *Aeromonas* species is very high, ranging from 96.7% to 100% [18,19].

Molecular identification methods using housekeeping genes, such as *gyrB* or *rpoD*, have been widely used for defining species and assessing the phylogenetic relationships within the genus *Aeromonas* [18,19,20,21]. While sequencing “housekeeping genes” can help in identifying *Aeromonas* species, it is not as reliable as whole-genome analysis for distinguishing closely related species [22,23]. Whole-genome sequencing provides more comprehensive data and is more effective for detailed taxonomic analyses.

Although our previously published data revealed some atypical biochemical features of the AB005 strain, phylogenetic analysis based on 16S rRNA and gyrB gene sequences confirmed its identity as *A. hydrophila* [24]. In the present study, whole-genome analysis was performed, showing that this strain represents a new species within the genus *Aeromonas*, for which the name *Aeromonas oralensis* AB005^T^ is suggested.

## 2. Materials and Methods

### 2.1. Sampling

Tissue samples were aseptically collected from the ulcerated muscle and gill lesions of seven diseased *Acipenser baerii* (average weight of 1080 ± 150 g, length of 56–68 cm) at a recirculating sturgeon farm in Oral, Western Kazakhstan (45°52′ N, 51°15′ E). Water in the recirculated aquaculture system was maintained at 22 °C, pH 7.0, and 7.3 mg/L dissolved O_2_ with 5–10% daily turnover through sand, mechanical, and biological filtration. The tissue samples were transported in oxygenated sterile vessels to Zhangir Khan West Kazakhstan Agrarian Technical University for bacteriological analysis [24].

### 2.2. 16S rRNA Gene Sequence Analysis

The isolated bacteria were cultured in Luria–Bertani (LB) medium for 24 h, and genomic DNA was extracted using a DNeasy PowerLyzer Microbial Kit (Qiagen, Hilden, Germany). The 16S rRNA gene was amplified by PCR using the universal bacterial primers 27F and 14292R, following established protocols [24]. The PCR products were sequenced using the BigDye Terminator v3.1 Cycle Sequencing Kit (PerkinElmer, Waltham, MA, USA) on an ABI PRISM capillary sequencer (Vaulx-en-Velin, France). Sequence reads were processed and corrected using the MEGA 11 software suite [25] and then merged using the EMBOSS package v6.6.0.0 [26]. The resulting 16S rRNA sequences were compared against the NCBI GenBank public database using BLASTN with default parameters (https://blast.ncbi.nlm.nih.gov/Blast.cgi, accessed on 28 February 2025). Multiple sequence alignments were generated using CLUSTAL X v2.1 [27], and phylogenetic trees were constructed using the maximum likelihood method in MEGA v11 [25]. Evolutionary distances were calculated using the Tamura–Nei correction model [28]. To evaluate the robustness of the phylogenetic tree, bootstrap analysis was performed with 1000 iterations [29]. Positions with missing data were excluded from the analysis. The 16S rRNA gene sequence of *Pseudaeromonas pectinilytica* AR1^T^ was used as the outgroup.

### 2.3. Genome Sequencing and Phylogenomic Analysis

Whole-genome sequencing was performed using the third-generation sequencing platform PromethION48 at the Center for Life Sciences, NLA, Nazarbayev University. The sequencing was performed on FLO-PRO002 flow cells using an SQK-LSK109 kit. Raw FAST5 signals were transformed into nucleotide sequence reads in the FASTQ file format using Guppy Basecaller v6.2.7 with a high-accuracy model [30]. Low-quality sequencing reads with a q-score below 9.0 were filtered out by Guppy during the base-calling process. A total of 15.07 Gb of bases were called and 3.54 million reads were generated. To eliminate residual adapters and barcodes, we ran Porechop v0.2.4 on the FASTQ files, followed by length filtering to discard reads that were shorter than 1 kb. The resulting clean dataset (~14.8 Gb in 3.2 million reads) was used for de novo assembly with Flye v2.9.3 [31] and annotated using RASTtk pipeline v2.0 [32].

The completeness of the *Aeromonas* AB005 genome was assessed using CheckM v1.0.18 [33]. The whole-genome shotgun project has been deposited in GenBank under the accession number CP187186.2. Genome-based phylogenetic analysis was conducted using the Type (Strain) Genome Server (TYGS) on the DSMZ platform (http://ggdc.dsmz.de/, accessed on 4 June 2024) [34]. Additionally, the Genome-to-Genome Distance Calculator (GGDC v3.0) and Ortho Average Nucleotide Identity (OrthoANI) with OAT v0.93.1 (OrthoANI Tool v0.93.1) [35] were used to assess the genomic similarity between the AB005 strain and closely related *Aeromonas-*type strains (Appendix A) [34]. Genome comparisons with the closest type strains were visualized using the Blast Ring Image Generator (BRIG v0.95) [36]. The OrthoVenn v3 web server was used to identify orthologous gene clusters, both unique and shared, between the AB005 strain and closely related *Aeromonas* type strains. The analysis was conducted using default parameters for the protein-coding genes of the strains [37]. Core- and pan-genome analyses were performed using the bacterial pan-genome analysis (BPGA v1.3) pipeline with the USEARCH v2.14.12 clustering algorithm, applying a 0.7 identity threshold [38]. Core- and pan-genome sizes were calculated based on up to 500 random unique combinations for each number of genomes (ranging from 1 to 4).

### 2.4. Assessment of Antimicrobial Resistance Gene Detection and Pathogenicity Prediction

The assembled genome of the *Aeromonas* isolate (AB005) was analyzed using the ResFinder v4.7.2 web tool [39,40] to identify acquired antimicrobial resistance (AMR) genes. Default settings were applied with a minimum identity threshold of 98%, and hits were reported along with their percentage identity, alignment length relative to gene length, and genomic location. In parallel, the genome was queried using the Resistance Gene Identifier (RGI v6.0.5) from the Comprehensive Antibiotic Resistance Database (CARD v4.0.1) [41] under strict criteria. The CARD output provided detailed annotations of the detected AMR determinants, including gene family, drug class, resistance mechanism, and match quality (percentage identity and length of the matching region). Furthermore, the predicted protein sequences translated from the assembled genome were submitted to the PathogenFinder v0.5.0 web server [42]. Using a minimum identity threshold of 92.555% and a Z-threshold of 3.0, the tool computed a prediction score and the probability of being a human pathogen and then enumerated the number of matched protein families (dividing them into pathogenic and non-pathogenic groups) while calculating the overall genome coverage by these matches.

## 3. Results

### 3.1. Phylogenetic Identification

A phylogenetic tree based on the 16S rRNA gene sequence was constructed to position the AB005 strain among the representative members of the genus *Aeromonas* (Figure 1). Maximum likelihood analysis of the 16S rRNA gene sequence (1403 bp) revealed 100.0% sequence identity between the AB005 strain and the type strain *Aeromonas hydrophila* ATCC 7966.

### 3.2. Genome Characteristics and Phylogenomic Analysis

The draft genome of the AB005 strain consists of 4,784,854 base pairs, with an average GC content of 61.2%. Annotation with Rapid Annotation using Subsystems Technology (RAST v2.0) identified 4607 protein-coding sequences, of which 349 were assigned to specific subsystems. In addition, 128 rRNA genes were identified in the genome (Table 1).

An analysis of the annotated genome of strain AB005 revealed that most of its genes were associated with various metabolic functions. These included genes involved in the metabolism of amino acid derivatives (378), carbohydrates (299), and proteins (265), as well as those involved in the biosynthesis of cofactors and secondary metabolites (179) and DNA metabolism (86). Additionally, genome annotation identified genes predicted to play a role in resistance to environmental stress. These included genes associated with oxidative stress (35), osmotic stress (7), detoxification (7), and periplasmic stress (6). This analysis also highlighted genes involved in numerous defense mechanisms, including those related to antibiotic and toxic compound resistance (36) and antibacterial peptide resistance (2). This suggests that the AB005 strain possesses robust adaptive capabilities enabling it to survive in challenging environmental conditions (Figure 2).

Pairwise phylogenomic comparison conducted using the TYGS server showed that the AB005 strain clustered with two other *Aeromonas* species. Its closest relative was identified as the type strain *Aeromonas hydrophila* ATCC 7966, with a digital DNA–DNA hybridization (dDDH) identity value of 68.9% (Figure 3). It also exhibited a close relationship with *A. hydrophila* subsp. *ranae* CIO 107985, with a dDDH value of 68.0%. Both dDDH values were below the recommended threshold of 70%, which was used to delineate the species. This indicates that the AB005 strain represents a distinct species within the genus *Aeromonas*.

The results of the OrthoANI-based phylogenomic tree and heat map differ from the dDDH identity values, indicating clustering of the AB005 strain with the *Aeromonas hydrophila* ATCC 7966^T^ species (Figure 4), with OrthoANI values ranging between 86.02 and 96.38%.

Comparative genomic analysis using OrthoVenn v3 revealed several unique gene clusters that distinguished the AB005 strain from closely related species, particularly *Aeromonas hydrophila* ATCC 7966^T^. In this analysis, nine unique orthologous gene clusters were identified. A closer examination of these clusters showed the presence of multiple paralogous genes, suggesting recent gene duplication events in AB005 (Figure 5). Orthologous gene clustering analysis across the four *Aeromonas* strains indicated a core-genome of 3231 clusters shared among all strains. Notably, AB005 harbored nine unique clusters that were absent in other strains, highlighting its distinct genomic content. Furthermore, AB005 shared 96 exclusive clusters with ATCC 7966^T^, indicating a moderate degree of relatedness. Overall, the identification of strain-specific clusters and the differences observed in the accessory genomes support the genomic divergence of AB005. These findings, consistent with the GGDC analyses, reinforce the classification of AB005 as a species within the genus *Aeromonas*.

A BLASTp analysis of the unique gene clusters revealed that most top hits corresponded to proteins from *Aeromonas veronii* and *Aeromonas hydrophila*, with high sequence identities (>95%) and extremely low E-values (as low as 0.0). These findings suggest that although the genes in AB005 are not entirely novel, they likely represent conserved regions or expanded gene families resulting from paralogous duplications. For instance, the clusters containing genes, such as cds-3796.peg.1044 and peg.1042, exhibited near-identical sequences, further supporting the occurrence of such duplications. These gene expansions may contribute to functional redundancy or diversification and potentially offer adaptive advantages under specific environmental conditions. Complementing these results, functional annotation of the unique gene clusters showed significant enrichment for their roles in transport (GO:0006810), metabolic processes (GO:0008152), and interspecies interactions (GO:0044419). These functions may underpin the adaptive mechanisms that allow AB005 to thrive in ecological niches distinct from those of *A. hydrophila*. Moreover, although many unique genes share homology with hypothetical proteins, the detection of conserved domains—such as VCBS domain-containing proteins and retention module-containing proteins—suggests potential roles in environmental adaptation, host interaction, and stress response. A collinearity analysis comparing the AB005 strain with *A. hydrophila* ATCC 7966^T^ revealed pronounced differences in genomic synteny (Figure 6).

Disrupted collinear blocks and rearranged genomic segments indicate structural variations, such as insertions, deletions, and inversions, which are unique to AB005 and further support its genomic distinctiveness, as well as its evolutionary divergence from *A. hydrophila*. Disruptions in synteny may affect gene regulation and expression, thereby contributing to phenotypic variation and adaptive potential. In addition, several other genes (for example, cds-3794.peg.2173 and peg.2866) are transposase-related, suggesting the involvement of mobile genetic elements that could have facilitated horizontal gene transfer and the acquisition of novel traits in AB005 (Figure 5). Moreover, the GO enrichment analysis highlighted a significant association with the positive regulation of viral transcription (GO:0050434), implying the presence of prophage elements or viral origin genes, which is consistent with the role of transposases in driving genomic diversity through horizontal gene transfer.

To assess the genomic diversity and evolutionary dynamics of *Aeromonas* AB005, core/pan-genome analysis was performed using BPGA v1.3 (Figure 7).

The core/pan-genome analysis indicated an open pan-genome, as the number of unique gene families continued to increase with the addition of new genomes. Meanwhile, the core-genome size declined, signifying substantial genetic variability within the studied *Aeromonas* strains. This pattern suggests high genomic plasticity, which is potentially driven by horizontal gene transfer or environmental adaptation. The observed genomic diversity, coupled with the identification of nine unique orthologous gene clusters from OrthoVenn v3, provides strong evidence that *Aeromonas* AB005 represents a distinct lineage within the *Aeromonas* genus.

The Blast Ring Image Generator (BRIG v0.95) was employed to conduct comparative genomic analysis between the AB005 strain and its closest *Aeromonas* relatives (Figure 8). This analysis provided a visual representation of genomic similarities and differences. The PHASTER pipeline (https://phaster.ca/, accessed on 1 June 2025) [47] identified three prophage elements within the genome of AB005, with lengths of 17.4 kb, 33.9 kb, and 13.2 kb (the sequence of the last prophage element was incomplete). Additionally, CRISPRCasFinder (CRISPRCasFinder. accessed on 5 June 2025) [48] detected nine CRISPR elements which likely contribute to the adaptive immunity of the strain against foreign genetic elements such as phages. Furthermore, the IslandViewer v4 web server was used to predict genomic islands (GIs), revealing 14 putative GIs within the genome (accessed on 29 October 2024) [41]. The GIs varied in size, ranging from 4212 bp (GI 4) to 15,187 bp (GI 5). GI 5, the largest genomic island, comprised 22 genes, of which 45.6% were annotated as hypothetical protein-coding genes. In contrast, GI 4, the smallest island, contained only four genes. A comparison of the GC content between these islands and the average GC content of the AB005 genome revealed distinct differences. All 14 GIs exhibited lower GC content, ranging from 43.8% to 58.0%, compared with the average GC content of the AB005 genome. This deviation suggests that potential horizontal gene transfer events might have contributed to the acquisition of these islands (Figure 8).

### 3.3. Antimicrobial Resistance Genes and Pathogenicity

The ResFinder v4.7.2 analysis identified two β-lactamase genes (*ampH* and *cphA1*/*imiH*) in the AB005 strain, with sequence identities of 96.23% and 93.73%, respectively, suggesting that AB005 is capable of inactivating β-lactam antibiotics such as ampicillin and amoxicillin. A complementary CARD analysis detected additional beta-lactamase variants (e.g., *cphA2* and *OXA-950*) with high identities (>96%), as well as an RND efflux pump gene (*rsmA*), supporting a multidrug resistance profile. In addition, PathogenFinder v0.5 returned a prediction score of 81.59 and estimated a 65.8% probability that AB005 is a human pathogen, with 51 protein family matches identified, among which 34 correspond to pathogenic families and 17 to non-pathogenic ones. Although the overall genome coverage by these matches is modest (0.88%), the skew toward pathogenic protein families strongly suggests that AB005 harbors virulence-associated factors (Figure 9).

The heat map summarizes the percentage identities and prediction metrics for a set of resistance genes and pathogenicity features across four *Aeromonas* strains: AB005, KN-Mc-6U21, ATCC 7966^T^, and DSM 7311^T^. The rows represent individual features derived from different analyses: Beta-lactam (ResFinder v4.7.2) denotes resistance genes identified using ResFinder v4.7.2; Beta-lactam (CARD) indicates resistance genes detected using the CARD database; Pathogenic Potential represents metrics derived from PathogenFinder v0.5., including human pathogen probability, prediction score, number of matches, and number of matched pathogenic versus non-pathogenic protein families. The color gradient from white to yellow to red indicates increasing values, with higher color intensity representing higher percentage identity or higher prediction scores. Cells displaying a value of 0.00 indicate that the corresponding feature was not detected in that strain. The left-hand-side annotation differentiates between the three feature categories: “Beta-lactam (ResFinder v4.7.2)” is displayed in sky blue, “Beta-lactam (CARD)” in navy, and “Pathogenic Potential” in red.

Importantly, a comparative heat map (Figure 1) integrating these data reveals distinct clustering patterns between AB005 and the reference strains (KN-Mc-6U21, ATCC 7966^T^, and DSM 7311^T^). In this heat map, the percentage identities of resistance genes from ResFinder v4.7.2 and CARD are shown alongside the pathogenicity metrics from PathogenFinder v0.5. The observed differences, including numerous 0.00 values, indicating the absence of specific resistance genes in the other strains, highlight that the AB005 strain exhibits a unique genomic profile. This distinct pattern implies that AB005 may represent a novel lineage with potentially different resistance and pathogenic traits compared with the other *Aeromonas* strains examined.

## 4. Discussion

Based on the 16S rRNA analysis, the isolate characterized in this study was initially classified as *Aeromonas hydrophila*. However, whole-genome analysis revealed that it represents a novel species within the *Aeromonas* genus. Biochemical tests, previously conducted on the AB005 strain [24], demonstrated positive results in several assays, including methyl red, oxidase, Voges–Proskauer, O-nitrophenyl-*β*-D-galactopyranoside (ONPG), and arginine dihydrolase assays. Additionally, this strain was found to be able to form acids from sucrose and trehalose. In contrast, it does not produce indole and tested negative for ornithine decarboxylase and D-xylose fermentation—differences that set it apart from typical *A. hydrophila* strains [12].

The comparative genomic analysis further underscores the distinctiveness of AB005. Notably, this analysis revealed several unique gene clusters that differentiate AB005 from closely related species, particularly *Aeromonas hydrophila* ATCC 7966^T^. An orthologous gene cluster analysis using OrthoVenn v3 identified nine unique clusters in the AB005 strain. Upon further examination, it was found that these clusters encompass multiple paralogous genes, suggesting recent gene duplication within the strain. The combined evidence from analyses of paralogous gene expansions, the presence of mobile genetic elements, distinct functional enrichments, and collinearity highlights the dynamic and unique genomic architecture of AB005. Although AB005 shows high sequence similarity to *A. hydrophila*, its unique genomic features and structural variations support its classification as a novel species within the genus.

Integrated analyses using ResFinder v4.7.2 and CARD further revealed that AB005 harbors a robust set of *β*-lactam resistance genes. For instance, ResFinder v4.7.2 detected two beta-lactamase genes (*ampH* and *cphA1*/*imiH*) with high percentage identities, whereas the CARD analysis identified additional variants (such as *cphA2* and *OXA-950*) along with an RND efflux pump gene (*rsmA*). Complementing these findings, PathogenFinder v0.5 returned a prediction score of 81.59 and a 65.8% probability that AB005 is a human pathogen, with a notable bias toward pathogenic protein family matches. These results collectively indicate that AB005 possesses both a comprehensive antimicrobial resistance repertoire and a higher pathogenic potential than the reference strains.

Furthermore, the core-genome and pan-genome analyses demonstrated that AB005 has an open pan-genome and a minimal core-genome, suggesting extensive genetic diversity. This dynamic genome structure likely contributes to the ecological adaptability and pathogenic potential of this strain.

Physiologically, AB005 is notable for its broad temperature tolerance (with optimal growth between 13 °C and 42 °C) and wide pH range (5–9). The strain tolerates salt concentrations between 0% and 3%, although higher levels (4% and above) significantly inhibit its growth [24]. These features further emphasize the adaptability of this strain to diverse environmental conditions.

Phylogenetic analyses using TYGS and GGDC confirmed that AB005 forms a distinct lineage within the *Aeromonas* genus. Despite sharing some sequence similarities with *A. hydrophila*, the unique combination of its genomic features, resistance gene profile, and physiological characteristics clearly distinguishes AB005 as a novel species.

## 5. Conclusions

Genomic analysis of the newly isolated *Aeromonas* strain AB005 revealed that it belonged to a distinct species within the *Aeromonas* genus. Pairwise phylogenomic comparison using the TYGS server showed that the AB005 strain clustered with *A. hydrophila* and *A. hydrophila* subsp. ranae, with dDDH identity values of 68.9% and 68.0%, respectively. Moreover, core-genome and pan-genome analyses, together with comprehensive resistance and pathogenicity profiling, provided strong evidence that the AB005 strain is distinct from previously characterized *Aeromonas* species. This unique genomic and phenotypic profile has important implications for clinical treatment and epidemiological tracking, suggesting that AB005 may exhibit virulence and resistance properties different from those of its close relatives.

## 6. Description of *Aeromonas oralensis* sp. nov.

*Aeromonas oralensis* (or. al. en. sis NL. Masc. adj. oralensis, referring to the city where it was isolated).

These bacterial cells are aerobic, motile, and Gram-negative rods. *A. oralensis* grows at temperatures between 13 and 42 °C and a wide pH range of 5–9, with an optimal temperature of approximately 25 °C and a pH of 6.5. This strain can tolerate salinity levels ranging from 0% to 3%, with its growth inhibited at higher concentrations, particularly at 5%. Positive results were observed in the methyl red, oxidase, Voges–Proskauer, ONPG, and arginine dihydrolase assays, and the strain was reported to form acids from sucrose and trehalose. Negative results were obtained for the production of ornithine decarboxylase, d-xylose, and indole. This strain was found to hydrolyze esculin and gelatin and produce hydrogen sulfide. The results of the lysine decarboxylase assay and acid formation from lactose varied. *A. oralensis* metabolizes carbohydrates via oxidative fermentation, as indicated by the O/F test. The draft genome sequence is 4,784,854 bp with an average GC content of 61.2%. The genome is available in GenBank (accession number CP187186.2), and the 16S rRNA gene sequence is registered under OK634406.1. The AB005 strain was isolated from diseased *Acipencer baerii* from the Oral region of Kazakhstan.

## Figures and Tables

**Figure 1 microorganisms-13-01680-f001:**
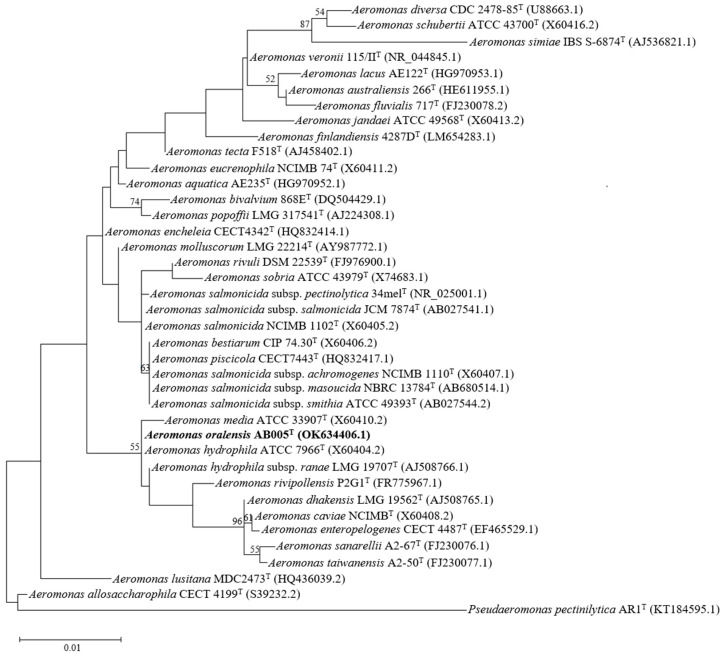
A phylogenetic maximum likelihood tree was constructed based on the 16S rRNA gene sequences of the *Aeromonas* strain AB005 (highlighted in bold) and representative members of the genus *Aeromonas*. Accession numbers are provided in brackets. Bootstrap values, expressed as percentages, are indicated at the nodes. The scale bar represents 0.01 changes per nucleotide position. *Pseudaeromonas pectinilytica* AR1^T^ is included as the outgroup.

**Figure 2 microorganisms-13-01680-f002:**
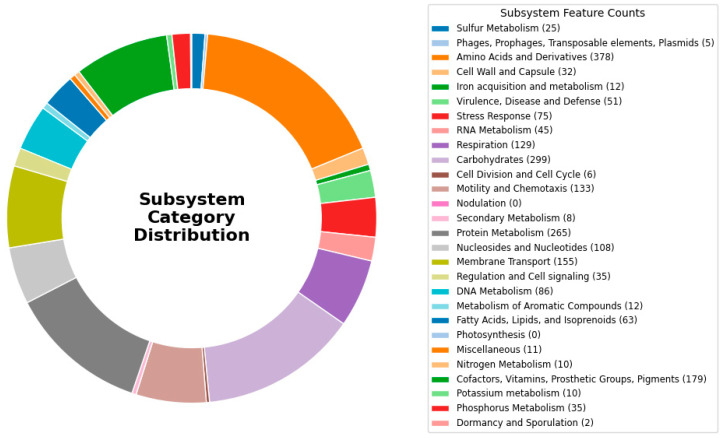
SEED subsystem category for the AB005 strain genome. Each colored bar corresponds to the number of genes allocated to each category.

**Figure 3 microorganisms-13-01680-f003:**
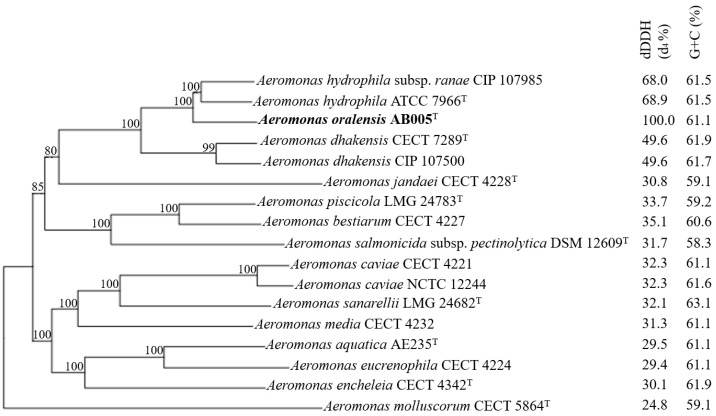
A phylogenomic tree of the *Aeromonas* strain AB005 and representative members of the genus *Aeromonas*. The tree was constructed using FastME 2.1.6.1 [44] from the Genome Blast Distance Phylogeny (GBDP). The branch lengths are scaled according to the GBDP distance formula d5. Numbers above the branches represent GBDP pseudo-bootstrap support values, calculated from 100 replications, with values greater than 80% indicating strong support. The average branch support is 97.5%. The tree is rooted at the midpoint [45] and visualized with PhyD3 [46]. The dDDH values (formula 2) and G + C contents are displayed on the right.

**Figure 4 microorganisms-13-01680-f004:**
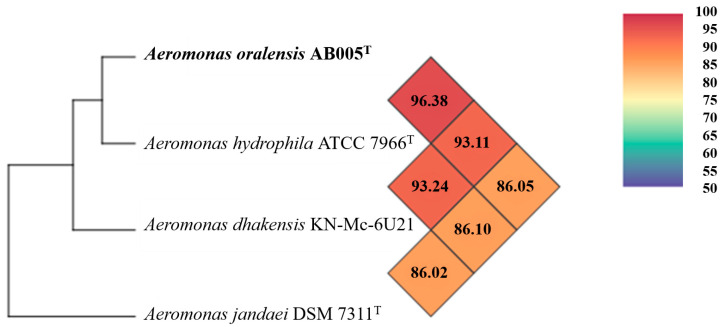
The heat map generated using OrthoANI values calculated using OAT software v0.93.1 for the AB005 strain and closely related *Aeromonas* species. Color codes represent the OrthoANI values in percentages.

**Figure 5 microorganisms-13-01680-f005:**
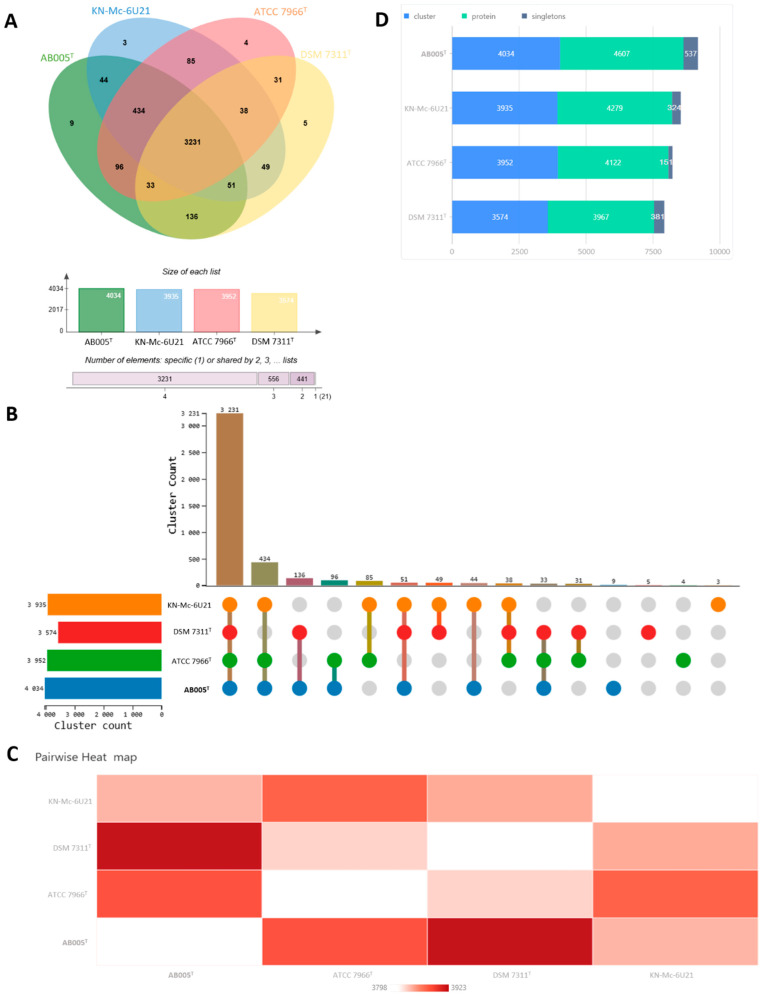
Orthologous gene clustering analysis between the AB005 strain and *Aeromonas dhakensis* KN-Mc-6U21, *Aeromonas hydrophila* ATCC 7966^T^, and *Aeromonas jandaei* DSM 7311^T^: (**A**) a classic Venn diagram illustrating the selected species; (**B**) an UpSet plot displaying the unique and shared orthologous clusters among the species; (**C**) a heat map showing the number of overlapping clusters between each pair of species; and (**D**) a bar chart representing the number of protein sequences, orthologous clusters, and singletons for each species.

**Figure 6 microorganisms-13-01680-f006:**
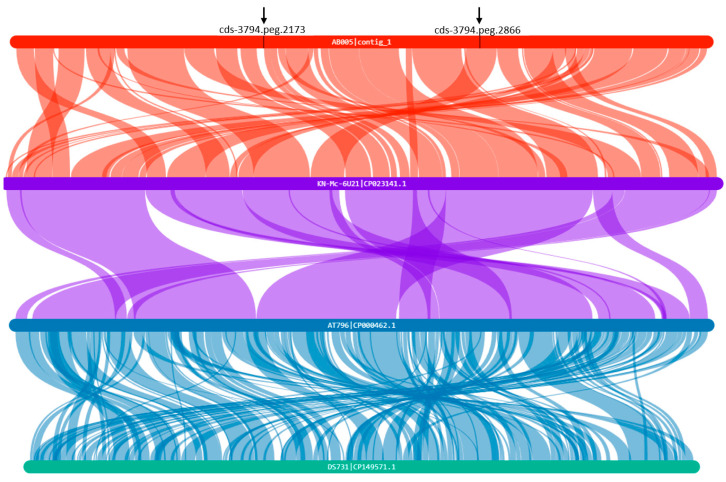
Collinearity analysis between the AB005 strain and *Aeromonas dhakensis* KN-Mc-6U21, *Aeromonas hydrophila* ATCC 7966^T^, and *Aeromonas jandaei* DSM 7311^T^. Black arrows indicate the locations of genes such as cds-3794.peg.2173 and peg.2866 on the chromosome of the AB005 strain.

**Figure 7 microorganisms-13-01680-f007:**
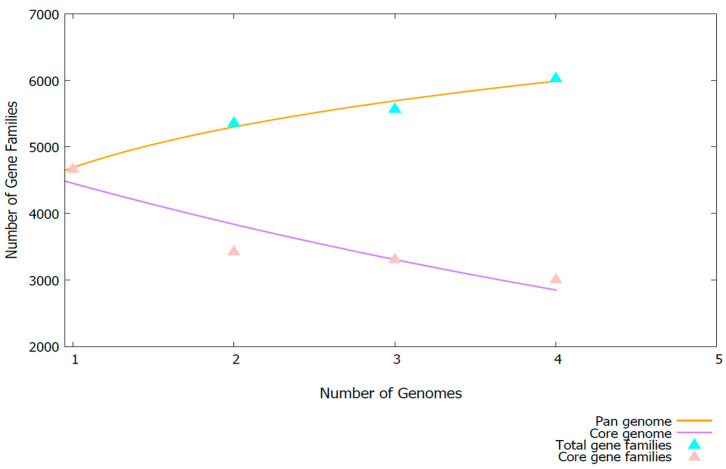
The core- and pan-genome development plot of *Aeromonas* AB005 in comparison with *Aeromonas dhakensis* KN-Mc-6U21, *Aeromonas hydrophila* ATCC 7966^T^, and *Aeromonas jandaei* DSM 7311^T^. The analysis was conducted using the BPGA v1.3 pipeline with the USEARCH v2.14.12 clustering algorithm at a 70% identity threshold. The core-genome (purple line) represents the set of conserved genes shared across all analyzed genomes, decreasing as more genomes are considered. The pan-genome (orange line) depicts the total number of gene families, increasing with additional genomes. Triangles represent the total gene families (cyan) and core gene families (pink) calculated from up to 500 unique random genome combinations.

**Figure 8 microorganisms-13-01680-f008:**
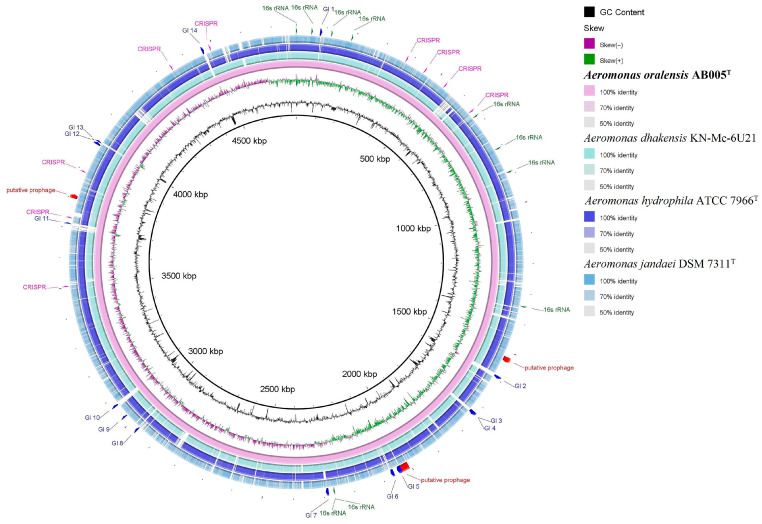
A circular representation of the genome of the *Aeromonas* strain AB005 compared with those of closely related *Aeromonas* species. The circular genome map was generated using the Blast Ring Image Generator (BRIG v0.95), a cross-platform tool [36]. The AB005 strain was utilized as the reference genome. Rings from inside to outside: G + C content (black); GC skew (−, purple; +, green); *Aeromonas* strain AB005 (pink); *Aeromonas dhakensis* KN-Mc-6U21 (pale blue); *Aeromonas hydrophila* ATCC 7966^T^ (blue); *Aeromonas jandaei* DSM 7311^T^ (turquoise). The positions of genomic islands (blue arcs), rRNA operons (green arcs), putative prophages (red), and CRISPR clusters (fuchsia) are also indicated. Genome sequence accession numbers: CP187186.2 (AB005^T^); CP023141.1 (KN-Mc-6U21); CP000462.1 (ATCC 7966^T^); CP149571.1 (DSM 7311^T^).

**Figure 9 microorganisms-13-01680-f009:**
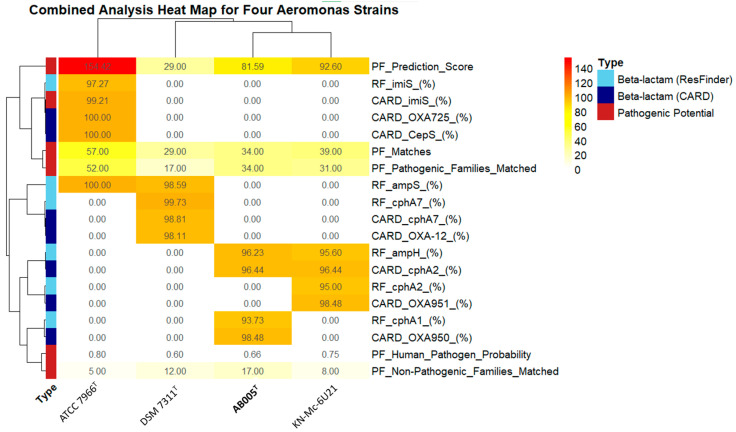
Comparative heat map of AMR and pathogenicity features in four *Aeromonas* strains.

**Table 1 microorganisms-13-01680-t001:** Genome statistics of closest *Aeromonas* genomes *.

Scientific Name	*Aeromonas oralensis* AB005^T^	*Aeromonas hydrophila* ATCC 7966^T^	*Aeromonas dhakensis* KN-Mc-6U21	*Aeromonas jandaei* DSM 7311^T^
Genome size (Mb)	4.7	4.7	4.9	4.6
G + C content (%)	61.2	61.5	61.5	59
N50 (Mb)	4.7	4.7	4.9	4.6
L50	1	1	1	1
Number of contigs	1	1	1	1
Number of coding sequences	4607	4161	4316	3967
Number of tRNAs	128	126	126	124
5S rRNA	11	11	11	11
16S rRNA	10	10	10	10
23S rRNA	10	10	10	10
NCBI Accession no.	CP187186.2	CP000462.1	CP023141.1	CP149571.1

* Data were retrieved from GenBank [43].

## Data Availability

The original contributions presented in this study are included in the article/Appendix A. Further inquiries can be directed to the corresponding author.

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
