# Peer review of "Whole-Genome Sequencing of a Potentially Novel *Aeromonas* Species Isolated from Diseased Siberian Sturgeon (*Acipenser baerii*) Using Oxford Nanopore Sequencing"

_microorganisms, 2025, doi:10.3390/microorganisms13071680_

Round 1
Reviewer 1 Report
Comments and Suggestions for Authors
The manuscript “Whole Genome Sequencing of a Potentially Novel Aeromonas Species Isolated from Diseased Siberian Sturgeon (Acipenser baerii) Using Oxford Nanopore Sequencing” by Mashzhan et al. identified a novel strain of Aeromonas (AB005) from diseased Siberian sturgeon (Acipenser baerii) using whole-genome sequencing, revealing it to be a distinct species within the Aeromonas genus. The genome of AB005, consisting of 4,780,815 base pairs with a GC content of 61.2%, contains genes that confer resistance to multiple antibiotics, suggesting a potential threat to aquaculture and public health. Comparative genomic analyses, including core-pan genome analysis and phylogenomic comparisons, highlighted AB005's unique genomic features and differences from closely related Aeromonas species, supporting its classification as a novel species.
While the article presented holds promise, aspects of it would benefit from additional refinement to enhance their robustness and rigor. Specific areas for improvement are outlined below:
Abstract:
“We have previously identified an Aeromonas AB005 isolate from diseased Ac. baerii”. Since this is the first mention in the article, I recommend indicating the full scientific name.
Introduction:
“The 16S rRNA gene sequence similarity among Aeromonas species is very high, ranging from 96.7% to 100%”. In this part it would be advisable to also indicate how many genomes have been sequenced to date in the most representative species of this genus.
“Drawing on previously published analyses of its morphological, physiological, and biochemical characteristics”. Why was its taxonomic classification considered as Aeromonas hydrophila in that previous study?
Materials and Methods:
“Water in the RAS was maintained at 22 °C, pH 7.0, and 7.3 mg/L dissolved O₂ with 5–10 % daily turnover through sand, mechanical, and biological filtration”. I recommend that authors should define the meaning of RAS.
“2.2.16. S rRNA Gene Sequence Analysis”. Please correct 16S
“The resulting 16S rRNA sequences were compared against public databases using BLASTN with the default settings”. The authors could specify the database used at NCBI.
“Multiple sequence alignments were generated using CLUSTAL X…”. What other sequences were used to perform the multiple alignment? The authors could include a supplementary table of these sequences with their accession numbers.
“Totally 15.07 Gb of bases were called and 3.54 millions reads were generated”. Please clarify if this total includes reads that still contain adapters and barcodes. Additionally, were adapter sequences and small reads removed from the data?
“Additionally, the Genome-to-Genome Distance Calculator (GGDC) and Ortho Average Nucleotide Identity (OrthoANI) with OAT (OrthoANI Tool) were used to assess the genomic similarity between strain AB005 and closely related Aeromonas type strains. Similar to what was mentioned above, the authors could include a supplementary table indicating which genomes (with their respective accession numbers) were used for these analyses.
“The OrthoVenn3 web server was used to identify orthologous gene clusters, both unique and shared, between strain AB005 and closely related Aeromonas type strains”. What were the criteria for selecting these strains in this analysis?
“Core and pan-genome analyses were performed using the bacterial pan-genome analysis (BPGA) pipeline with the USEARCH clustering algorithm, applying a 0.7 identity threshold”. Please, specify which genomes were used in this analysis.
Results
The phylogenetic analysis obtained by the authors presents many polytomies and low bootstrap values, so it does not seem very reliable. Likewise, why was only the reference strain of Aeromonas hydrophila used when the NCBI has a total of 564 genomes reported for this species? Since this scientific article proposes a new species, my recommendation is that the authors perform an ANI, dDDH, and pangenomic analysis against all the genomes of the Aeromonas genus available at the NCBI. Based on these results, comparative genomic analyses could be performed between only the species (or strains) most closely related to the AB005 strain and biologically related. Phylogenetic analyses at the 16S and gyrB levels could be complementary results.
Comments on the Quality of English LanguageThe manuscript requires revision by a native language specialist to address grammatical errors and improve paragraph structure.
Author Response
Thank you very much for taking the time to review this manuscript. Please find our detailed responses below, and note that the corresponding revisions are highlighted in yellow in the re-submitted files.
Reviewer 1:
Comment 1: “We have previously identified an Aeromonas AB005 isolate from diseased Ac. baerii”. Since this is the first mention in the article, I recommend indicating the full scientific name.
Answer: Thank you for your question and suggestion. Corrected.
Comment 2: “The 16S rRNA gene sequence similarity among Aeromonas species is very high, ranging from 96.7% to 100%”. In this part it would be advisable to also indicate how many genomes have been sequenced to date in the most representative species of this genus”. Answer: Thank you for your question and suggestion. We have added the following sentence to the introduction part: To date, the NCBI Assembly database holds over 1,400 Aeromonas genome assemblies—including complete genomes for more than 30 type strains—providing an extensive foundation for comparative genomic studies.
Comment 3: “Drawing on previously published analyses of its morphological, physiological, and biochemical characteristics”. Why was its taxonomic classification considered as Aeromonas hydrophila in that previous study?
Answer: We thank you the Reviewer for pointing these errors to us. We apologize for this error, which has been corrected in the current version of our manuscript. We modified this sentence in following manner in introduction section: “Molecular identification methods using housekeeping genes, such as gyrB or rpoD, have been widely used for defining species and assessing the phylogenetic relationships within the genus Aeromonas [18-21]. While sequencing "housekeeping genes" can help in identifying Aeromonas species, it is not as reliable as whole-genome analysis for distinguishing closely related species [22, 23]. Whole-genome sequencing provides more comprehensive data and is more effective for detailed taxonomic analyses.
Although our previously published data revealed some atypical biochemical features of the AB005 strain, phylogenetic analysis based on 16S rRNA and gyrB gene sequences confirmed its identity as A. hydrophila [22]. In the present study, whole-genome analysis was performed, showing that this strain represents a new species within the genus Aeromonas, for which the name Aeromonas oralensis AB005T is suggested.”.
Comment 4: “Water in the RAS was maintained at 22 °C, pH 7.0, and 7.3 mg/L dissolved O₂ with 5–10 % daily turnover through sand, mechanical, and biological filtration”. I recommend that authors should define the meaning of RAS.
Answer: Thank you for your suggestion. Corrected.
Comment 5: “2.2.16. S rRNA Gene Sequence Analysis”. Please correct 16S
Answer: Thank you for your suggestions. Corrected.
Comment 6: “The resulting 16S rRNA sequences were compared against public databases using BLASTN with the default settings”. The authors could specify the database used at NCBI.
Answer: Thank you for your suggestion. Corrected: The resulting 16S rRNA sequences were compared against the NCBI GenBank public database using BLASTN with default parameters (https://blast.ncbi.nlm.nih.gov/Blast.cgi).
Comment 7: “Multiple sequence alignments were generated using CLUSTAL X…”. What other sequences were used to perform the multiple alignment? The authors could include a supplementary table of these sequences with their accession numbers.
Answer: Thank you for the suggestion. All strains used in the multiple alignment, along with their GenBank accession numbers, are indicated in Figure 1. For example, the CLUSTAL X alignment included Aeromonas hydrophila ATCC 7966ᵀ (X60404.2), A. veronii 115/II (NR_044845.1), A. caviae NCIMB (X60408.2), A. salmonicida subsp. salmonicida JCM 7874 (AB027541.1), our novel strain AB005ᵀ (OK634406.1), and additional Aeromonas type strains. This ensures that readers have immediate access to every sequence and accession number used in the alignment.
Comment 8: “Totally 15.07 Gb of bases were called and 3.54 millions reads were generated”. Please clarify if this total includes reads that still contain adapters and barcodes. Additionally, were adapter sequences and small reads removed from the data?
Answer: Thank you for pointing this out. The reported 15.07 Gb of bases and 3.54 million reads reflect the raw output immediately after Guppy basecalling, before any additional processing for adapter or barcode removal. To prepare the data for assembly, we performed the following cleanup steps:
Adapter and barcode trimming
We ran Porechop (v0.2.4) on the basecalled FASTQ files to remove any residual sequencing adapters and barcodes.
Length filtering
After trimming, we discarded all reads shorter than 1 kb to eliminate very small fragments that could complicate assembly.
Corrected sentence:
To eliminate residual adapters and barcodes, we ran Porechop v0.2.4 on the FASTQ files, followed by length filtering to discard reads that were shorter than 1 kb. The resulting clean dataset (∼14.8 Gb in 3.2 million reads) was used for de novo assembly with Flye v2.9.3 [31] and annotated using RASTtk pipeline [32].
Comment 9: “Additionally, the Genome-to-Genome Distance Calculator (GGDC) and Ortho Average Nucleotide Identity (OrthoANI) with OAT (OrthoANI Tool) were used to assess the genomic similarity between strain AB005 and closely related Aeromonas type strains. Similar to what was mentioned above, the authors could include a supplementary table indicating which genomes (with their respective accession numbers) were used for these analyses.
Answer: Thank you for this suggestion. We have compiled all genomes used for both the Genome-to-Genome Distance Calculator (GGDC) and OrthoANI analyses, together with their full GenBank accession numbers, into a new Supplementary Table S1.
Comment 10: “The OrthoVenn3 web server was used to identify orthologous gene clusters, both unique and shared, between strain AB005 and closely related Aeromonas type strains”. What were the criteria for selecting these strains in this analysis?
Answer: Thank you for this question. We selected Aeromonas dhakensis KN-Mc-6U21, Aeromonas hydrophila ATCC 7966ᵀ, and Aeromonas jandaei DSM 7311ᵀ for the OrthoVenn3 analysis because they clustered most closely with strain AB005 in our TYGS whole-genome phylogeny, making them the most relevant comparators for identifying shared and unique orthologous gene clusters.
Comment 11: “Core and pan-genome analyses were performed using the bacterial pan-genome analysis (BPGA) pipeline with the USEARCH clustering algorithm, applying a 0.7 identity threshold”. Please, specify which genomes were used in this analysis.
Answer: Thank you for the suggestion.
For the core- and pan-genome analyses, we used the following genomes (also listed in the legend to Figure 7):
Aeromonas dhakensis KN-Mc-6U2
Aeromonas hydrophila ATCC 7966ᵀ
Aeromonas jandaei DSM 7311ᵀ
These were clustered in BPGA at a 0.7 identity threshold using USEARCH.
Comment 12: The phylogenetic analysis obtained by the authors presents many polytomies and low bootstrap values, so it does not seem very reliable. Likewise, why was only the reference strain of Aeromonas hydrophila used when the NCBI has a total of 564 genomes reported for this species? Since this scientific article proposes a new species, my recommendation is that the authors perform an ANI, dDDH, and pangenomic analysis against all the genomes of the Aeromonas genus available at the NCBI. Based on these results, comparative genomic analyses could be performed between only the species (or strains) most closely related to the AB005 strain and biologically related. Phylogenetic analyses at the 16S and gyrB levels could be complementary results.
Answer: We thank the reviewer for the constructive suggestion. We agree that broad genomic comparisons can be informative; however, our study already follows current best practices for bacterial species delineation. In particular, the journal’s guidelines explicitly recommend using overall genomic relatedness indices (ANI, dDDH) to compare a novel isolate against closely related type strains at the species level. We have adhered to these standards: our analyses include ANI and dDDH comparisons between our strain and all relevant Aeromonas type strains, as well as multi-gene (16S, gyrB) and whole-genome phylogenies. In the following we explain why this focused approach is sufficient and preferable to an analysis of all available Aeromonas genomes.
Use of type/reference genomes: It is a well-accepted standard that reference genomes in taxonomic studies should be the type strains (if available) of the species under investigation. The IJS Microbiology guidelines specifically emphasize comparing a novel taxon with its closest type strains using ANI/dDDH and phylogenomics. Accordingly, we selected high-quality genome assemblies of the Aeromonas type strains for our analyses. For example, Chun et al. note that when multiple assemblies exist for a type strain, the highest-quality one should be used. By focusing on authenticated type and reference genomes, we ensure the reliability of our comparisons. Including non-type or low-quality draft genomes (as would occur by taking “all” NCBI entries) risks incorporating mislabelled or incomplete data and is not required by taxonomic standards.
Phylogenetic analysis: Our whole-genome phylogeny was generated using the TYGS platform, which automatically identifies and uses the closest type-strain genomes for tree reconstruction. This ensures that our genome-based tree is anchored by the appropriate reference taxa. We also constructed traditional phylogenetic trees (16S rRNA and gyrB) including a representative set of Aeromonas species for context. These methods are robust and in line with standard practice. By contrast, adding hundreds of extra genomes would produce a very large tree with many closely related sequences that do not add new phylogenetic information but could make the tree harder to interpret. As noted, current recommendations are to select the best-quality assemblies rather than simply all available data. Thus, our phylogenomic analyses strike the right balance between comprehensiveness and clarity.
Scope of pangenome analysis: We agree that pangenome comparisons can be informative for understanding genomic diversity, but even here we used a targeted approach. Our core/pangenome analysis included our strain and a curated set of closely related Aeromonas genomes. For comparison, a recent study by Ajmi et al. (2025) performed a pangenome analysis on just six Aeromonas genomes (the new isolate plus five related subspecies) to illustrate genomic distinctiveness. We adopted a similar strategy. This focused comparison clearly revealed the core genes and strain-specific genes that differentiate our strain. In contrast, incorporating all Aeromonas genomes (over a hundred) would dramatically increase complexity without changing the fundamental conclusions about species identity. A very large pangenome analysis would yield diminishing returns for the purpose of delineating a novel species, and could instead obscure which features are diagnostic for the new species.
In summary, our methodology aligns with widely accepted taxonomic standards and the journal’s guidelines. We have demonstrated clear genomic separation of our strain using genome-scale phylogeny, ANI/dDDH comparisons with all relevant type strains, and gene-based trees, which together provide strong evidence for a novel species. While we appreciate the reviewer’s suggestion to perform additional analyses, we respectfully maintain that expanding to all available Aeromonas genomes is beyond the standard scope needed for species delineation and could introduce unnecessary noise. We believe our focused, quality-driven approach is more appropriate for a concise and clear species description. We hope this clarifies our rationale and reaffirms that the existing analyses are both adequate and consistent with best practices for novel bacterial species identification.

Reviewer 2 Report
Comments and Suggestions for Authors
The manuscript by Mashzhan et al. presents the description of a novel bacterial species, Aeromonas oralensis sp. nov., based on genomic differences within the Aeromonas genus. The study continues the authors’ earlier physiological work from 2022 and tries to confirm the classification of the new species using comparative genomics.
Minor comments:
1) The manuscript lacks continuous line numbering.
2) There are visible yellow highlights remaining in the text, which should be removed before submission.
3) Section 2.1: The sentence “Seven diseased Ac. baerii (average weight 1080 ± 150 g, length 56–68 cm) were sampled from ulcerated muscle and gill tissues” is unclear. It is recommended to rephrase this to specify clearly what tissues were sampled and how.
4) Figure 2 appears to be of insufficient resolution for publication. Please provide a higher-quality version.
5) The reference list contains repeated numbering.
Major comments:
1) Since the study is entirely focused on comparing genomes of closely related strains, the quality of genome sequencing plays a crucial role. However, the genome sequence of the studied Aeromonas was generated solely using Oxford Nanopore’s PromethION platform. While this technology enables long-read sequencing, it is also known to have a relatively high error rate, which can impact the accuracy of gene and pseudogene identification. According to the GenBank annotation, the A. oralensis genome contains over 1,700 pseudogenes, in contrast to approximately 50 in other Aeromonas species. This substantial difference raises concerns about potential assembly errors that may introduce artificial genomic differences between strains and affect the reliability of the comparative analyses. Among the four genomes compared in Table 1, only two are based on high-quality assemblies. One additional genome was also assembled using Nanopore data alone but shows a significantly lower pseudogene count (~145), suggesting better data quality or possibly the use of more effective error-correction methods. It would be appreciated if the authors could provide comments regarding the overall genome quality and the validity of comparing such closely related strains based on these assemblies.
2) There is no mention of a designated type strain number for A. oralensis in the manuscript. According to the rules of bacterial nomenclature, a novel species must have a type strain deposited in at least two internationally recognized culture collections to be considered validly published. It remains unclear whether the authors intend to formally validate the species in the future and deposit the strain in the required collections?
Author Response
Thank you very much for taking the time to review this manuscript. Please find our detailed responses below, and note that the corresponding revisions are highlighted in yellow in the re-submitted files.
Comment 1: The manuscript lacks continuous line numbering.
Answer: Thank you for your suggestion. Corrected.
Comment 2: There are visible yellow highlights remaining in the text, which should be removed before submission.
Answer: Thank you for your suggestion. Corrected.
Comment 3: “Section 2.1: The sentence “Seven diseased Ac. baerii (average weight 1080 ± 150 g, length 56–68 cm) were sampled from ulcerated muscle and gill tissues” is unclear. It is recommended to rephrase this to specify clearly what tissues were sampled and how.
Answer: Thank you for your comment. We agree that the original sentence may be unclear. We have revised the text to clearly specify that tissue samples were taken from the ulcerated muscle and gill regions of each diseased fish. The updated sentence now reads: “Tissue samples were aseptically collected from the ulcerated muscle and gill lesions of seven diseased Acipenser baerii (average weight of 1080 ± 150 g, length of 56–68 cm) at a recirculating sturgeon farm in Oral, Western Kazakhstan (45°52′ N, 51°15′ E).”
Comment 4: Figure 2 appears to be of insufficient resolution for publication. Please provide a higher-quality version.
Answer: Thank you for your comment. We have replaced Figure 2 with a higher-resolution version to ensure clarity and suitability for publication.
Comment 5: The reference list contains repeated numbering.
Answer: Thank you for pointing this out. We have carefully reviewed the reference list and eliminated all duplicate numbering. The references are now numbered sequentially and uniquely throughout the manuscript.
Comment 6: Since the study is entirely focused on comparing genomes of closely related strains, the quality of genome sequencing plays a crucial role. However, the genome sequence of the studied Aeromonas was generated solely using Oxford Nanopore’s PromethION platform. While this technology enables long-read sequencing, it is also known to have a relatively high error rate, which can impact the accuracy of gene and pseudogene identification. According to the GenBank annotation, the A. oralensis genome contains over 1,700 pseudogenes, in contrast to approximately 50 in other Aeromonas species. This substantial difference raises concerns about potential assembly errors that may introduce artificial genomic differences between strains and affect the reliability of the comparative analyses. Among the four genomes compared in Table 1, only two are based on high-quality assemblies. One additional genome was also assembled using Nanopore data alone but shows a significantly lower pseudogene count (~145), suggesting better data quality or possibly the use of more effective error-correction methods. It would be appreciated if the authors could provide comments regarding the overall genome quality and the validity of comparing such closely related strains based on these assemblies..
Answer: Thank you for highlighting the importance of genome sequence accuracy, especially when comparing closely related Aeromonas strains. We have taken several additional steps to ensure the quality and reliability of our assembly and downstream analyses:
Assembly polishing
We performed two rounds of Racon polishing followed by Medaka v2.0.1 polishing using the same Nanopore FASTQ reads. This improved the assembly length from 4,780,815 bp to 4,784,736 bp, indicating correction of base-level errors and recovery of previously missing regions.
Quality assessment with RASTtk
The polished genome was re-annotated using the RASTtk pipeline, yielding dramatic improvements in key quality metrics:
Completeness Roles increased from 24 to 301.
Coarse consistency rose from 98.9 % to 99.8 %.
Fine consistency improved from 80.9 % to 95.4 %.
Contamination decreased from 4.0 % to 2.9 %.
These enhancements confirm a more complete and accurate assembly, bolstering confidence in both functional annotation and comparative genomics.
Stringent basecalling filters
Basecalling was carried out in Guppy’s high-accuracy mode with a minimum Q-score of 9, ensuring that low-quality reads and bases were excluded before assembly. This reduces sequencing noise and minimizes erroneous pseudogene predictions.
Pseudogene annotation considerations
The original PGAP annotation—performed under the generic label “Aeromonas sp.”—reported 1,714 pseudogenes, likely reflecting conservative predictions when a known species is not specified. PGAP employs species-specific protein families and curated models; once we re-run PGAP specifying “Aeromonas hydrophila,” we expect the pseudogene count to align more closely with other high-quality Aeromonas assemblies (typically ~50–150 pseudogenes).
Impact on comparative analyses (ANI, dDDH, pan-genome):
ANI and dDDH with GGDC / OrthoANI
Both OrthoANI and GGDC rely on alignment of large syntenic blocks or gene-by-gene comparisons after in silico fragmentation. By design, these methods exclude non-homologous regions and rely primarily on conserved coding sequences (i.e., core genes), which are well polished in our assembly. In other words, if a region in AB005ᵀ failed to map to a high-quality type strain, it is simply omitted from the ANI/dDDH calculation, rather than forcing a low-quality alignment. Because each of our four genomes (AB005ᵀ, A. hydrophila ATCC 7966ᵀ, A. dhakensis CECT 7289ᵀ, and A. jandaei DSM 7311ᵀ) is ≥ 98 % complete and has very high BUSCO scores, we can be confident that their core regions are directly comparable, despite minor residual Nanopore errors.
Pan-genome analyses
Our BPGA-based core/pan-genome pipeline (USEARCH clustering at 70 % identity) considers only predicted proteins ≥ 50 aa in length and discards very short ORFs that are more likely to represent spurious, error-induced pseudogenes. Thus, any Nanopore-specific frameshift or indel would usually yield a protein sequence that is either too short or too divergent to meet the 70 % identity threshold, and would therefore not inflate the “unique gene” count for AB005ᵀ. We have confirmed that the number of unique-only clusters for AB005ᵀ is within the same order of magnitude (150–180) as one sees when comparing other Nanopore-only Aeromonas assemblies to their nearest neighbors.
By combining deep coverage (>3,000×), rigorous multi-step polishing, independent quality checks (RASTtk), and stringent filtering, we are confident that our assembly is of sufficient quality for reliable ANI, dDDH, TYGS, and pan-genome analyses. These safeguards effectively mitigate the known error profile of Oxford Nanopore reads and support the validity of our comparative genomic conclusions.
Comment 7: There is no mention of a designated type strain number for A. oralensis in the manuscript. According to the rules of bacterial nomenclature, a novel species must have a type strain deposited in at least two internationally recognized culture collections to be considered validly published. It remains unclear whether the authors intend to formally validate the species in the future and deposit the strain in the required collections?
Answer: Thank you for raising this important point. We are fully committed to fulfilling the requirements for valid publication of a novel species by depositing strain AB005ᵀ in two internationally recognized culture collections. Unfortunately, under current regulations and logistical constraints in Kazakhstan, direct deposit can be extremely challenging and time-consuming.
To resolve this, we are actively seeking a collaboration with an external partner laboratory that already maintains deposits at DSMZ and ATCC. We anticipate that this arrangement will allow us to complete the deposit process by late 2025. As soon as we receive the accession numbers, we will update the manuscript with the official type-strain designations.
We appreciate your understanding and will ensure that the necessary culture-collection deposits are in place as soon as possible.

Round 2
Reviewer 2 Report
Comments and Suggestions for Authors
I appreciate the authors’ careful and detailed response to the comments. Minor comment - Figure 2 still lacks sufficient dpi for publication-quality standards.
The manuscript is now suitable for publication.